# Shortcuts to adiabaticity for open systems in circuit quantum electrodynamics

Zelong Yin[1], Chunzhen Li[1], Jonathan Allcock[1], Yicong Zheng [1], Xiu Gu [1], Maochun Dai[1], Shengyu Zhang[1] & Shuoming An [1✉]

Shortcuts to adiabaticity are powerful quantum control methods, allowing quick evolution into target states of otherwise slow adiabatic dynamics. Such methods have widespread applications in quantum technologies, and various shortcuts to adiabaticity protocols have been demonstrated in closed systems. However, realizing shortcuts to adiabaticity for open quantum systems has presented a challenge due to the complex controls in existing proposals. Here, we present the experimental demonstration of shortcuts to adiabaticity for open quantum systems, using a superconducting circuit quantum electrodynamics system. By applying a counterdiabatic driving pulse, we reduce the adiabatic evolution time of a single lossy mode from 800 ns to 100 ns. In addition, we propose and implement an optimal control protocol to achieve fast and qubit-unconditional equilibrium of multiple lossy modes. Our results pave the way for precise time-domain control of open quantum systems and have potential applications in designing fast open-system protocols of physical and interdisciplinary interest, such as accelerating bioengineering and chemical reaction dynamics.

[1] Tencent Quantum Laboratory, Tencent, 518057 Shenzhen, Guangdong, China. ✉email: shuomingan@tencent.com

A diabatic processes – which preserve the non-degenerate instantaneous eigenstates of time-dependent Hamiltonians – have important applications in quantum technologies, including quantum simulation[1,2] and computation[3,4]. Though adiabatic evolution is, in principle, relatively robust against parameter fluctuations and environmental noise[5], in the noisy intermediate-scale quantum era, decoherence is still an obstacle preventing its widespread application. Shortcut to adiabaticity (STA) addresses this issue by finding fast trajectories that connect the initial and final states of slow-paced adiabatic protocols. Since STA was first proposed[6], it has found many applications, including atom cooling, trapped atom and ion transportation[7,8], spin population transfer[9–12], implementing quantum logic gates[13,14] and quantum thermodynamics[15]. Due to freedom in choosing intermediate trajectories, time-dependent control parameters of a system can be adjusted in different ways, resulting in various STA protocols[16]. In particular, the method of counterdiabatic (CD) driving[17,18] adds an auxiliary control $H_{CD}$ to the reference Hamiltonian to suppress unwanted diabatic transitions. This adiabatic-following feature makes CD driving robust to parameter errors[8] and suitable for fast holonomic gates[19] and efficient quantum heat engines[20].

While STA finds widespread applications in closed quantum systems, its generalisation to open quantum systems is of fundamental interest. There are two strategies for this generalisation: first, one can stick with STA designed for closed systems and mitigate environmental effects by utilising redundant degrees of freedom[12]; second, one can directly attempt to accelerate open system adiabatic dynamics. For open classical systems, a swift-equilibration protocol similar to STA was used to accelerate the equilibration of a Brownian particle[21]. Recently, this idea was extended to the field of biology to guide the probability distribution of genotypes in a population along a specified path and time interval[22]. For open quantum systems, CD driving can be designed theoretically based on non-Hermitian Hamiltonians[23,24] or Lindblad dynamics[25]. However, it remains challenging to conduct such experiments, as they often require complex controls such as engineered system-bath interactions[26].

Here, we show the generalisation of STA to an open circuit QED (cQED) system consisting of multiple dissipative bosonic modes. When the time-dependent controls are varied sufficiently slowly[27], the system evolves adiabatically within its time-dependent decoherence-free subspace (DFS)[28]. Analogously to the CD driving for closed quantum systems, we deduce the diabatic part of the Liouvillian, which causes non-equilibrium transitions, and add a unitary control to counteract it. Consequently, we can enforce fast adiabatic evolution of the time-dependent DFS, i.e., a system initialised in the DFS remains so at all times[29]. However, when multiple lossy modes are present – as is common in many experimental scenarios – under the one-port driving of our setup (see below), only one hybrid mode at a time can be under perfect CD control. To realise STA in a multi-mode setting, we further develop an analytical, open-loop control protocol, which we term multi-mode optimal control (MMOC). In MMOC, we make use of non-adiabatic dynamics during ringup(ringdown) to achieve the desired final equilibrium in a duration much less than that required by a slow varying adiabatic reference process; see below. Utilising redundant degrees of freedom, we can also minimise a user-defined merit function, which we choose here to be the maximum driving amplitude to avoid undesired qubit excitations[30]. See Fig. 1a for a schematic overview of the system dynamics under the CD and MMOC procedures.

## Results

### Open circuit QED system.
Motivated by potential quantum computing applications such as improved qubit readout cycle time

and fidelity, we choose an experimental setup (Fig. 1b) consisting of two coupled resonator modes[31], $a$ and $b$, with coupling strength $J/2\pi \approx 10.5$ MHz. Mode $b$ is dispersively coupled to a transmon qubit[32] and mode $a$ is coupled to a feedline at an effective temperature of 75 mK with strength $\kappa_a/2\pi \approx 11.4$ MHz, which serves as a cold bath. Given the dispersive coupling to the transmon qubit, the coupled modes $a$ and $b$ can be decoupled into qubit-dependent hybrid modes (normal modes) $a'^{0,1}$ and $b'^{0,1}$ (see Methods). A coherent drive with amplitude $\epsilon(t)$ at frequency $\omega_d/2\pi = 6.44025$ GHz is generated by an arbitrary waveform generator and applied through the feedline. The system dynamics is inferred via time-traced output homodyne detection using the input-output theory as detailed in Supplementary Note 1.

**Open system STA by CD driving.** We first implement CD driving designed for lossy hybrid mode $b'$ to achieve a fast ringup in a target duration $t_f$, with the qubit kept in the ground state. For a chosen reference drive $\epsilon(t)$, the required additional CD control can be derived from the Lindblad dynamics or time-dependent DFS of the system (see Supplementary Note 2 and 3). The CD driving in the rotating frame of driving frequency $\omega_d$, including the reference $\epsilon(t)$ and the auxiliary control, takes the form:

$$\epsilon_{CD}(t) = \epsilon(t) - i\frac{\dot{\epsilon}(t)}{\Delta_r^0 - i\kappa_r^0/2}, \tag{1}$$

where $\Delta_r^0 \equiv \omega_r^0 - \omega_d$ and $\kappa_r^0$ are the frequency detuning and decay constant respectively, for the hybrid resonator mode $b'^0$ and qubit state in $|0\rangle$. For convenience, the superscripts are omitted in what follows. Note that in the dissipation-free limit $\kappa_r \to 0$, the closed system CD solution is readily recovered[8]. We characterize the performance of the CD driving in terms of the quantum speed limit for open quantum systems[33] and conclude that the CD driving has optimal quantum efficiency within the space of all available pulses (see Supplementary Note 4).

As shown in Fig. 2, we apply the reference driving $\epsilon(t)$ with a $\sin^2$-shape ringup, i.e. $\epsilon(t) = \epsilon_0 \sin^2(\pi t/2t_f)$ before $t_f$ and $\epsilon(t) = \epsilon_0$ afterwards. This $\sin^2$-shape reference waveform is chosen to give a smooth, hardware-friendly CD driving pulse. For both $t_f = 30$ ns and 100 ns, we compare the performance of the reference driving $\epsilon(t)$ and CD driving $\epsilon_{CD}(t)$. To demonstrate a speedup compared with the adiabatic timescale[34], estimated to be on the order of $\kappa_r^{-1}$ (see Supplementary Note 3), we also apply an adiabatic $\sin^2$ driving. Until $t_f = 800$ ns $\gg \kappa_r^{-1}$, we observe a relatively good $\sin^2$-shape response, indicating adiabatic evolution. Our results show an equilibrium time of 100 ns for the CD driving and 350 ns $\sim 5\kappa_r^{-1}$ for a quench driving. For $t_f = 30$ ns, the large and rapidly varying CD pulse induces out-of-equilibrium excitation of the untargeted mode $a'$. As a consequence, $a'$ requires an extra relaxation time of 75 ns $\sim 5\kappa_f^{-1}$ to return to its equilibrium. For $t_f = 100$ ns, equilibrium is achieved almost immediately after $t_f$ and the non-equilibrium excitation of $a'$ is negligible. Since the sampled output signal is a coherent superposition of the input driving field and the leakage of the system (see Supplementary Note 1 for details), the spiked IQ trajectory during the CD driving in Fig. 2c does not imply non-equilibrium dynamics. Additional data, corresponding to different $t_f$ and a comparison between open and closed system CD drives, can be found in Supplementary Note 9.

The CD drive Eq. (1) can only accelerate adiabatic dynamics for a single, qubit-state-dependent hybrid mode at a time and may excite other modes out of their instantaneous eigenstates. This issue can be mitigated with a relatively small driving-detuning ratio for the untargeted modes (Fig. 1c). However, due to the time-energy uncertainty, when the protocol duration is

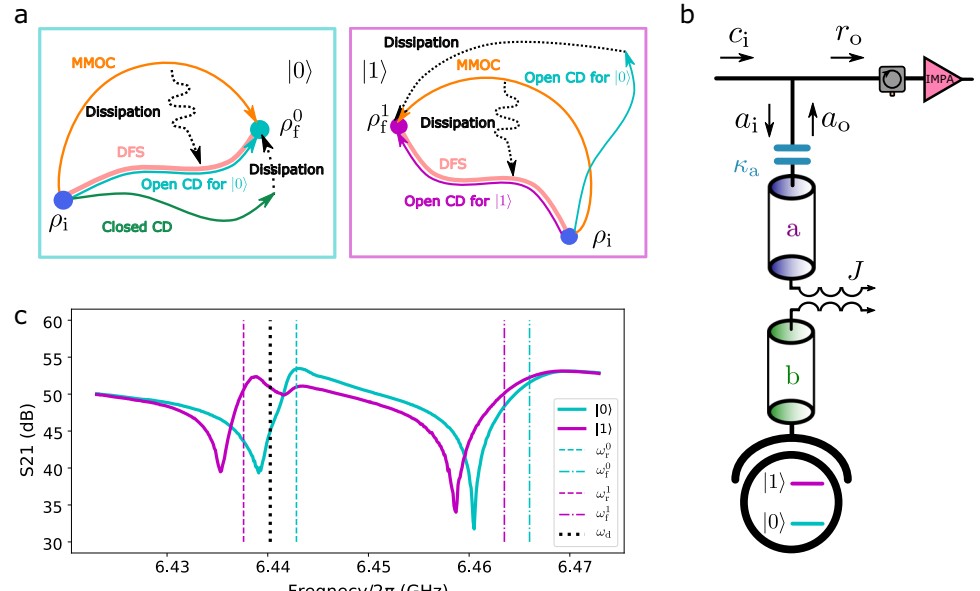

**Fig. 1 Experimental setup and principles of shortcut to adiabaticity (STA). a** Schematic of relevant dynamics. An adiabatic protocol transfers the initial state $\rho_i$ to the final state $\rho_f^{0,1}$ along the light pink trajectories in the decoherence-free subspace (DFS), which is precise only in the infinite time limit. To accelerate this adiabatic trajectory, a counterdiabatic (CD) driving can be applied to cancel non-adiabatic transitions, making the adiabatic approximation exact. Due to the effect of dissipation, CD driving for closed quantum systems has limited effectiveness, while its open system extension achieves arbitrary speed up for a single-mode conditioned on the qubit state. To realise STA for multiple modes and independent of qubit states, we develop the multi-mode optimal control (MMOC) protocol, which includes non-adiabatic trajectories at intermediate times and reaches both final states $\rho_f^{0,1}$ at a given target time. **b** Setup of the superconducting circuit. The resonator mode $b$ is dispersively coupled to a transmon qubit with strength $g/2\pi \approx 80$ MHz. The Purcell filter mode $a$ is coupled to the resonator mode with strength $J/2\pi = 10.5$ MHz and to the feedline with strength $\kappa_a/2\pi = 11.4$ MHz. From residue qubit excitation, the temperature of the environment is measured to be 75 mK, justifying the cold-bath approximation of the feedline. Driving pulses are applied through the input field $c_i$, and the output field $r_O$ is amplified by an impedance-matched parametric amplifier (IMPA) and homodyne detected to infer the system dynamics. **c** Measured transmission spectrum $S_{21}$ and hybrid-mode frequencies for both qubit states. The driving frequency $\omega_d/2\pi = 6.44025$ GHz is allocated roughly in the middle of $\omega_r^0/2\pi = 6.4427$ GHz and $\omega_r^1/2\pi = 6.4378$ GHz, where the 0 or 1 in the upper right corner denotes the qubit state. The significant shift of resonant dips is partly due to the nonlinearity of the cavity when it is driven with large power. See Supplementary Note 1 for more details on the asymmetry and frequency shifts.

reduced, the correspondingly larger drive amplitudes mean such unwanted excitations cannot be avoided entirely.

**Multi-mode optimal control**. To eliminate the excitation of the untargeted mode $a'$ and remove the qubit-state dependence of the driving, we propose the following MMOC protocol. The MMOC pulse is based on a multi-section ansatz: the protocol duration is divided into $m$ equal-spaced sections, each with a constant complex amplitude, i.e. $\epsilon(t_{j-1} < t < t_j) = \epsilon_j, j \in \{1, 2, 3, ..., m\}$. During the ringup stage, the control pulse is subject to the boundary conditions $\epsilon(0) = 0$ and $\epsilon(t_f) = \epsilon_0$. During the reset stage, the boundary conditions are reversed. To equilibrate four hybrid modes in time $t_f$, we utilise the underlying Langevin dynamics and obtain four linear equilibrium-transfer equations connecting the pulse vector $\boldsymbol{\epsilon}$ and the equilibrium difference $\mathbf{y}$ with a transfer matrix $\mathbf{G}$ as $\mathbf{y} = \mathbf{G} \cdot \boldsymbol{\epsilon}$. If the complex hybrid detuning $\widetilde{\Delta} \equiv \Delta - i\kappa/2$ is found for each hybrid mode, $\mathbf{G}$ and $\mathbf{y}$ can be analytically derived. These equilibrium-transfer equations can be solved via singular value decomposition of $\mathbf{G}$ as long as $m \geq 4$. The resulting $\boldsymbol{\epsilon}$ takes the form $\boldsymbol{\epsilon} = \boldsymbol{\epsilon}_e + \sum_{i=1}^{m-4} x_i \boldsymbol{\epsilon}_i$, where the essential vector $\boldsymbol{\epsilon}_e$ can be analytically solved for, and $\boldsymbol{\epsilon}_i$ are $m - 4$ vectors orthogonal to $\boldsymbol{\epsilon}_e$. See Supplementary Note 5 for a detailed derivation. The extra degrees of freedom $x_i$ are chosen numerically to minimise the maximum amplitude component of $\boldsymbol{\epsilon}$. As a result, we can obtain a 5 dB reduction in the maximum amplitude compared with the essential vector. See Supplementary Note 6 for more discussion on the performance and speed limit of this protocol.

Figure 3 shows our results for the fast equilibration of all four lossy hybrid modes of the coupled oscillator system. Choosing $m = 10$, we first apply a $t_f = 60$ ns ringup pulse for fast system equilibration, and then transfer the system to the vacuum state at the end with another 60 ns reset pulse. According to the time-traced IQ trajectories (Fig. 3b, c and insets), different modes undergo different non-equilibrium dynamics and end up in their respective equilibrium states after the MMOC pulse. Our results show that fast unconditional multi-mode ringup and depletion are almost achieved at the desired time and reduce the duration required to achieve equilibrium compared with natural relaxation. The accelerated depletion shows the advantage MMOC has over a simple rectangular pulse which is turned off at the end.

One issue preventing our protocols from further acceleration is the Kerr nonlinearity correction term $\frac{1}{2}K(b'^\dagger)^2 b'^2$ to the resonator mode dispersive Hamiltonian[35]. While our methods are designed to work in the linear regime, increasing the protocol speed requires fast-growing driving power, and accounting for nonlinearity becomes essential. However, in the weakly nonlinear regime, i.e., where driving power is well below the first-order dissipative phase transition point[36], we empirically – in both simulation and experiment – find the effect of nonlinearity can be largely mitigated by including a mean resonator frequency shift $\sim Kn$[37] in our protocols, where $n$ is the equilibrium photon number in the resonator mode. If the drive exceeds this dissipative phase transition point, not only does the microwave output suddenly increase, we also observe the transmon qubit becoming excited in a qualitatively similar way to a previously reported result[30]. This result is shown in Fig. 4, where we stimulate

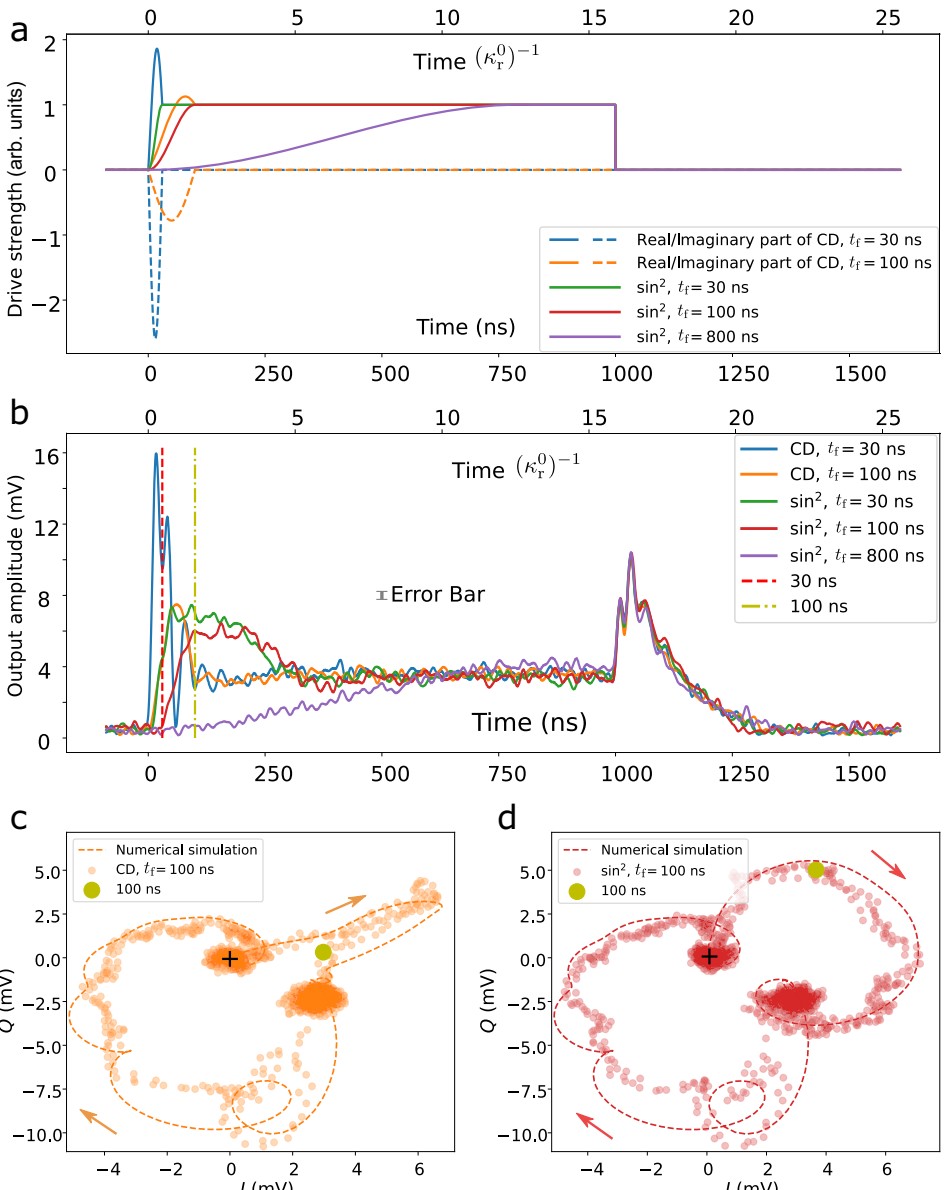

**Fig. 2 Open system shortcut to adiabaticity by counterdiabatic (CD) driving and bare drivings. a** Different pulses used in the experiment. The $\sin^2$ ringups with final time $t_f = 30$ ns (green) and 100 ns (red) are used, along with their corresponding CD drivings (blue and orange). The drivings are kept constant at $\epsilon_O$ after $t_f$. A slow varying $\sin^2$ driving with $t_f = 800$ ns (purple) is used to illustrate the adiabatic ringup timescale. **b** Output amplitude (in mV) measured by room-temperature FPGA. End times of the CD driving protocols are marked for $t_f = 30$ ns (red) and 100 ns (yellow). The non-equilibrium excitation after $t_f = 30$ ns is caused by excitation of the Purcell filter mode due to the relatively small resonator-filter detuning. The error bar is the standard deviation of the points in the equilibrium states at 990 ns. **c, d** We decompose the output signal into its (I)n-phase and (Q)uadrature components, and plot the corresponding IQ trajectories for the $t_f = 100$ ns $\sin^2$ driving (**d**) and its CD driving (**c**). Trajectories begin at the points indicated by black crosses. Arrows alongside the simulation indicate the direction of the time evolution. The simulated trajectories are calculated based on the input-output formalism detailed in Supplementary Note 1. To avoid the difficulty of direct tomography, we can infer the bosonic system mean-field state from the output signal. The good fit between simulation and experiment data suggests that the CD driving does indeed realise adiabatic following. The mismatch mainly comes from the weak nonlinearity of the resonator. All experimental points are averaged over $3 \times 10^4$ experiments. A Savitzky-Golay filter with window length 21 and polynomial order 3 is applied to improve the signal-noise ratio.

the resonator mode with different amplitudes and durations and measure the remaining ground state population. Our results support the theory[38] that the resonator phase transition and the qubit excitation coincide.

## Discussion

We have experimentally extended STA to an open quantum system. Our CD method accelerates adiabatic dynamics of the

DFS of a single driven-dissipative bosonic mode to occur within 100 ns, compared with the 800 ns of the regular slow-varying scheme. Furthermore, by utilising possible non-adiabatic dynamics, our MMOC protocol – based on methods from optimal control theory – can achieve unconditional adiabatic dynamics for multiple lossy bosonic modes simultaneously in a similar duration. It is worth pointing out that the ringdown pulse cannot be obtained by simply reversing the ringup pulse as can be done in the closed system. Time-reversal symmetry is broken

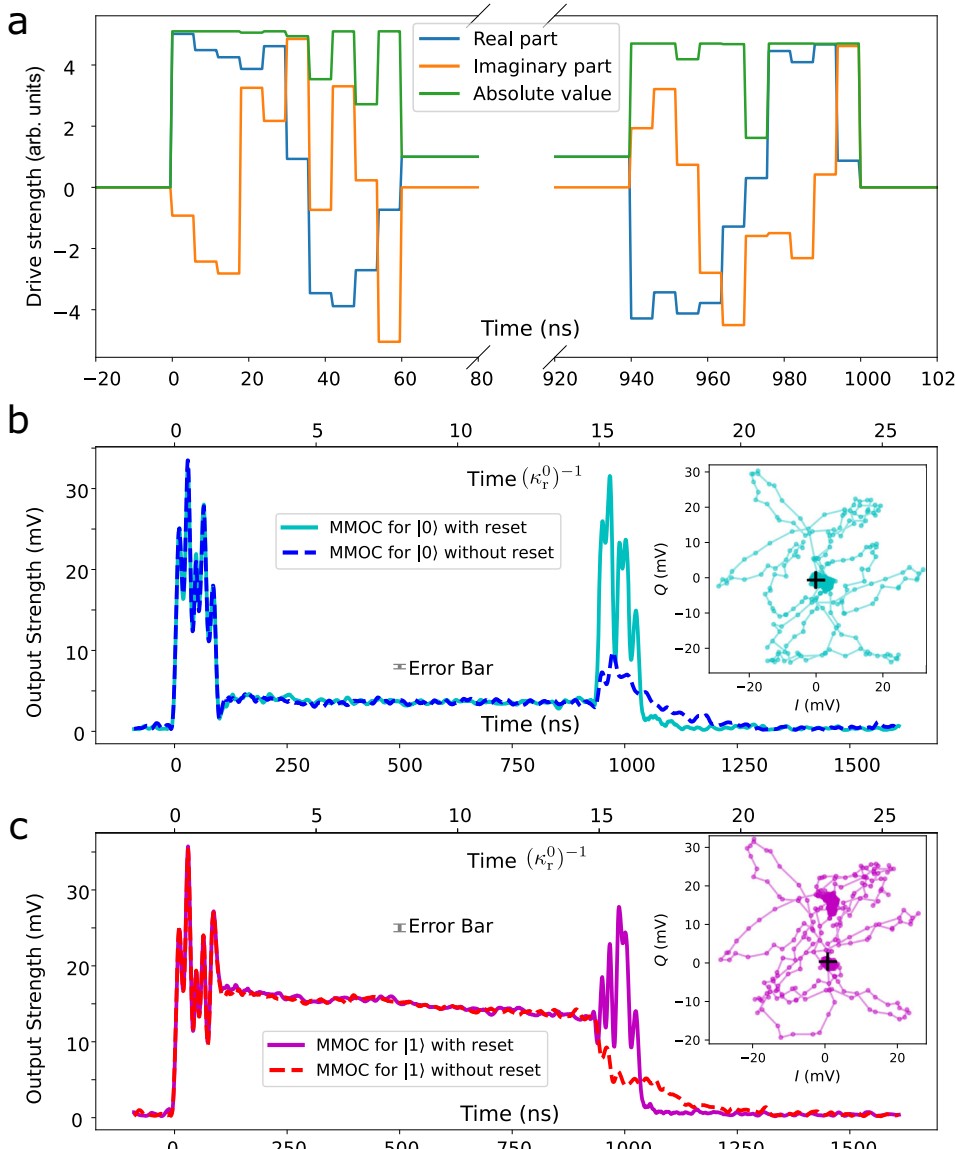

**Fig. 3 Multi-mode optimal control (MMOC) by single-port driving. a** MMOC pulses. A multi-section pulse with target equilibrium time $t_f = 60$ ns is applied to shortcut thermal equilibrium of both resonator and filter modes for different qubit states simultaneously. A different pulse is used to reset to the vacuum state in the end. **b, c** Measured output amplitudes (in mV) for the MMOC pulses with and without reset. The steady output signal is achieved for qubit state $|0\rangle$ (**b**) and $|1\rangle$ (**c**) about 30 ns later than the target time, likely due to the high driving amplitude and low-Q energy-storing components in the feedline, such as the impedance-matched Josephson parametric amplifier. This explanation is reinforced by the observation of a similar 30 ns tail with a far-detuned and high-amplitude driving which, in principle, will not excite the multi-mode cQED system (Supplementary Note 9). In **c**, the steady output decays over time due to the $T_1$ decay of the qubit. The error bar is the standard deviation of the points in the equilibrium states at 930 ns. The good final reset performance for the mixed qubit states demonstrates that MMOC works for both qubit states simultaneously. The corresponding IQ trajectories are plotted inset, from which we can infer the system undergoes highly non-equilibrium dynamics during the MMOC pulse. Trajectories begin at points indicated by black crosses. All data are processed in the same way as in Fig. 2.

under open-system dynamics. Compared with other ansatz-based numerical optimal control methods which can be more prone to getting stuck in local minima, our MMOC approach is based on a strong initial analytical solution that can be further optimised based on experimental data to mitigate error arising from initial parameters, nonlinearity and control distortions.

The methods we have presented here are a first step to accelerating open system dynamics in a broader range of settings. The open system CD suppresses diabatic transitions out of the DFS and opens the door for STA to more applications, such as quantum thermodynamics[39], reservoir engineering[40] and holonomic quantum computation based on bosonic codes[41].

The open-loop MMOC protocol, being independent of qubit-state, is suitable for fast resonator ringup(ringdown)[42,43] and can be used to reduce measurement cycle times in quantum error correction protocols. We also test its impact on qubit readout fidelity by applying MMOC to the rising edge of the readout pulse. Results are given in Supplementary Note 9, and show that MMOC can improve readout fidelity of the $|0\rangle$ state. For the $|1\rangle$ state, T1 decay occurring during the MMOC pulse currently leads to a slightly decreased readout fidelity, but this can be addressed either through an improved T1, or better statistical analysis of the readout data[44] (experiment in progress). Both methods can also potentially be applied to other dissipative

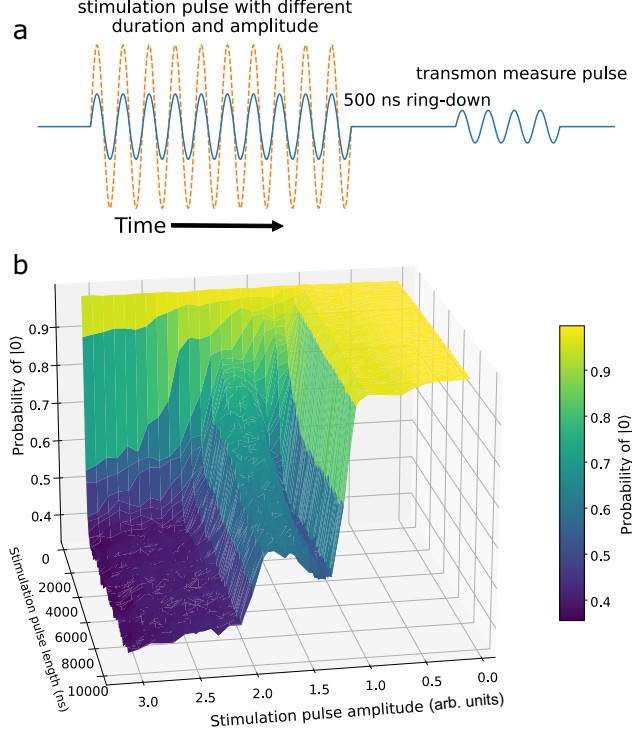

**Fig. 4 Qubit excitation by feedline driving. a** The pulse sequence used to check the impact of the microwave driving from the feedline on the transmon qubit. We apply a stimulating microwave pulse with a fixed frequency $\omega_d$ and different amplitudes and durations. We wait for 500 ns to clear the resonator-filter population before performing the qubit dispersive measurement. **b** Population of $|0\rangle$ as a function of the amplitude and duration of the stimulating pulse. Once a critical pulse amplitude is exceeded, the qubit is excited. Our results are qualitatively similar to previous findings[30]. To avoid unwanted excitation, we optimise the driving pulse amplitude and the speed of the protocol. For other linear lossy bosonic mode systems without a dispersively coupled qubit (e.g. optomechanics), STA will not be limited in this way.

bosonic systems, such as optical resonant cavities and optomechanics[45–47]. It is worth pointing out that, by setting the decay rate $\kappa = 0$, MMOC can be recast as an approach to closed system STA, with applications in bus-mediated gates in experimental quantum computing platforms[48,49]. As recent theoretical developments have shown, open-system STA also has interdisciplinary applications such as accelerating biophysics[50], bioengineering[22] and chemical reaction dynamics[51].

## Methods

**Experimental setup and calibration.** Our devices consists of a tunable transmon qubit coupled to a $\lambda/4$ microwave resonator with coupling $g/2\pi \approx 80$ MHz. The resonator is coupled to an individual $\lambda/4$ Purcell filter with coupling $J/2\pi \approx 10$ MHz. The hybrid mode frequencies are $\omega_r^0/2\pi = 6.4427$ GHz, $\omega_r^1/2\pi = 6.4378$ GHz and $\omega_f^0/2\pi = 6.4634$ GHz, $\omega_f^1/2\pi = 6.4673$ GHz and linewidths $\kappa_r^0 = 1/62.88$ ns, $\kappa_r^1 = 1/77.93$ ns and $\kappa_f^0 = 1/17.86$ ns, $\kappa_f^1 = 1/15.64$ ns. The frequencies $\omega_r, \omega_f$ can be estimated by fitting the transmission $S_{21}$ shown in Fig. 1c with the input-output theory of Supplementary Note 1, or by fitting the measured time-traced IQ plots with numerical trajectories obtained from quantum Langevin dynamics. The hybrid linewidths $\kappa_r, \kappa_f$ are obtained by measuring the decay time constant of the quenched ringdown. All spectrum parameters are characterised to a precision of $<2\pi \times 0.1$ MHz.

Driving pulses $\epsilon(t)$ are generated by an arbitrary waveform generator with a sampling rate of 2 GS/s and up-converted to the driving frequency by IQ mixing with the LO microwave signal. The readout signal is first amplified with an impedance-matched Josephson parametric amplifier, then by high electron mobility transistors, and finally by room temperature amplifiers. It is further homodyne demodulated, digitised by an analogue-to-digital converter at 1 GS/s

and analysed by a room-temperature DAQ FPGA. Each experimental point in the figures is an average of $3 \times 10^4$ experiments.

**Hybrid-mode dynamics.** Here we show how the interacting dissipative modes are decoupled so that CD driving is realised for a single lossy hybrid mode. The derivation is based on a simplified version of the circuit model shown in Fig. 1b, where the input gate capacitor is ignored, and the qubit's effect is implicitly accounted for through a state-dependent shift on the filter and resonator modes. The exact input-output relations based on the entire circuit are derived in Supplementary Note 1, which are equivalent to the results of this simplified model by reparametrization.

In the dispersive regime of cQED[52], after applying the rotating wave approximation, the bare system Hamiltonian in the driving frame (at frequency $\omega_d$) reads

$$H^{0,1} = \Delta_a a^\dagger a + \Delta_b^{0,1} b^\dagger b + J(a^\dagger b + b^\dagger a), \qquad (2)$$

where $a, b$ are bare filter and resonator modes. The superscripts denote the qubit state, $\Delta_{a(b)} \equiv \omega_{a(b)} - \omega_d$ is the driving detuning of mode $a(b)$, $J$ is the coupling strength between the modes, and the qubit-induced dispersive shift is $2\chi = \Delta_b^1 - \Delta_b^0$. Here we have set $\hbar = 1$. Since our protocol time is much shorter than the qubit lifetime, decoherence is dominated by relaxation $\kappa_a$ through the cold bath at an effective temperature of 75 mK. In the Heisenberg picture, the system dynamics according to the input-output theory is[53]

$$\dot{a} = -i\Delta_a a - \frac{\kappa_a}{2} a - iJb - \sqrt{\kappa_a} a_i, \qquad (3)$$

$$\dot{b} = -i\Delta_b^{0,1} b - iJa, \qquad (4)$$

where $a_i$ is the time-dependent input field. A linear transformation of $a$ and $b$ decouples these equations, resulting in the hybrid modes $a'^{0,1}, b'^{0,1}$ with frequencies $\Delta_f^{0,1}, \Delta_r^{0,1}$ and linewidths $\kappa_f^{0,1}, \kappa_r^{0,1}$. The hybrid modes are defined so that $[a'^i, a'^{i\dagger}] = [b'^i, b'^{i\dagger}] = 1$ for $i = 0, 1$, whereas $[a'^i, b'^i] \neq 0$ due to environmental effects. Omitting the superscripts for convenience, we have

$$\dot{a}' = -i\Delta_f a' - \frac{\kappa_f}{2} a' - c_f \sqrt{\kappa_a} a_i, \qquad (5)$$

$$\dot{b}' = -i\Delta_r b' - \frac{\kappa_r}{2} b' - c_r \sqrt{\kappa_a} a_i, \qquad (6)$$

where $c_f, c_r$ are coefficients from the linear transformation. Equation (6) is equivalent to the master equation in the Schrodinger picture[54]:

$$\dot{\rho}_{b'} = -i[H_r(t), \rho_{b'}] + \kappa_r \mathcal{D}(b')\rho_{b'}, \qquad (7)$$

where $H_r(t) = \Delta_r b'^\dagger b' - (ic_r \sqrt{\kappa_a} a_i(t) b'^\dagger + h.c.)$. For coherent driving fields, we can approximate the Hamiltonian as $H_r(t) = \Delta_r b'^\dagger b' - (i\epsilon_r(t) b'^\dagger + h.c.)$, where the effective drive is $\epsilon_r(t) = c_r \sqrt{\kappa_a} \langle a_i(t) \rangle$. Based on the master Eq. (7), we can derive (see Supplementary Note 3) the CD driving Eq. (1) for a single hybrid mode.

## Data availability
Source data are provided with this paper in the Source data file.

## Code availability
Codes used in the theoretical calculation are available from the corresponding author on reasonable request.

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

## Acknowledgements

We thank J.N. Zhang, C.Y. Hsieh, Z.Q. Yin, Z.B. Yang, G.H. Huang and X. Chen for helpful discussions and comments. We thank the electronics team of Tencent Quantum Lab for preparing the room-temperature electronics.

## Author contributions

Z.L.Y., S.M.A. and C.Z.L. developed the theory. S.M.A. and Z.L.Y. performed the experiment. J.A. edited the manuscript. S.M.A., Z.L.Y., Y.Z., X.G. and M.D. analysed the data. S.Z. supervised the project. All authors contributed to the writing of the manuscript.

## Competing interests

The authors declare no competing interests.
