## [Peer Review File · Nature Communications]

Reviewers' Comments:

Reviewer #1:

Remarks to the Author:

In the present work, the authors apply the method of shortcuts to adiabaticity in an open system composed by two coupled quantum oscillators and a two-level system. They theoretically derive the shortcut control which can drive one of the oscillators (one mode actually) to a nonzero equilibrium state without exciting the other oscillator or the two-level system, in a much faster time than the usual slow adiabatic method. They also find the control which brings quickly the oscillator back to the vacuum state. This method can be applied only for two coupled modes. In order to extend it to the case with more coupled modes, the authors use optimal control, in order to design the control inputs, which can quickly achieve the same tasks as before. The simulation predictions for the system are experimentally confirmed using a circuit quantum electrodynamics system.

We believe that the present work is significant for the field of quantum dynamics and control because it presents very interesting theoretical methods which are also experimentally applied in circuit quantum electrodynamics, a promising platform for quantum computation. The article is well written and scientifically sound, while the methods employed are explained in detail in the supplementary material. For these reasons we believe that the article warrants publication in Nature Communications, after the authors take into account the following observations:

1. They should explain what are the IQ initials appearing in the IQ trajectories displayed in several figures.

2. The authors need to cite some related works. Specifically, the following article where numerical optimal control is used to find inputs in order to cool a noisy (open) quantum harmonic oscillator, and in the long time limit the system can be driven also along a decoherence free path:

D. Stefanatos, Optimal efficiency of a noisy quantum heat engine, Phys. Rev. E 90, 012119 (2014).

Also, some articles where optimal control is used to obtain piecewise constant pulses, like those in the paper, which can be used to cool one of two coupled quantum oscillators:

X. Wang, S. Vinjanampathy, F. W. Strauch, and K. Jacobs, Ultraefficient cooling of resonators: beating sideband cooling with quantum control, Phys. Rev. Lett. 107, 177204 (2011).

S. Machnes, J. Cerrillo, M. Aspelmeyer, W. Wieczorek, M. B. Plenio, and A. Retzker, Pulsed laser cooling for cavity optomechanical resonators, Phys. Rev. Lett. 108, 153601 (2012).

D. Stefanatos, Fast cavity optomechanical cooling, Automatica 73, 71-75 (2016).

3. Line 13, there is an unnecessary "in" after "including". Also, in Ref. [15] it is "Jarzynski".

Reviewer #2:

Remarks to the Author:

In their manuscript "Shortcuts to Adiabaticity for Open Systems in Circuit Quantum Electrodynamics" the authors experimentally demonstrate a shortcut to adiabaticity for a system consisting of two transmon qubits and an oscillator. They propose a technique to perform the shortcut for an open quantum system as opposed to a closed system. They address an important experimental problem of designing STA protocols that can simultaneously suppress diabatic transitions in the presence of multiple modes. The multi-mode suppression is achieved using optimal control theory approach.

I found the paper mostly well written, and the experiments to be well done. However, in order to be of interest to the broad audience of Nature Communications, I believe significant improvements

to the paper should be done. For example, demonstrating that the proposed STA protocol can be used to improve qubit readout fidelity could be one way of adding value. In its current form, the results seem incomplete and would be more suitable for a specialized journal.

Comments:

1. Fig. 4 is only introduced in the discussion section. The related discussion should be either moved to the results section or the supplementary material.
2. In order to be able to assess the performance of the presented MMOC approach, it would have been interesting to see it compared to other known strategies for optimizing resonator pulse shapes, such as CLEAR demonstrated in ref [41] of the paper or just a rectangular pulse. What are the benefits of the presented scheme over other numerical optimization schemes? Is the optimized MMOC approach still a shortcut to adiabaticity? I would think that the adiabatic trajectory is no longer followed.
3. From the narrative it is not entirely clear why is the presence of the qubit important for the experiment. The application for readout optimization is only mentioned in the very end of the paper. This should be clarified. Similarly, in the abstract applications such as biophysics are given as an example, but the performed experiment seems more related to qubit readout optimization.
4. How significant impact did the inclusion of κ into the counterdiabatic driving protocol have? It would be interesting to see if the counterdiabatic driving designed for open quantum system works significantly better than the approximation derived for closed system.
5. Font size in the figures is quite small. I would suggest increasing them for better readability.

Point-by-point response

Referees' original comments are in black. Our responses are in blue.

Referee #1:

In the present work, the authors apply the method of shortcuts to adiabaticity in an open system composed by two coupled quantum oscillators and a two-level system. They theoretically derive the shortcut control which can drive one of the oscillators (one mode actually) to a nonzero equilibrium state without exciting the other oscillator or the two-level system, in a much faster time than the usual slow adiabatic method. They also find the control which brings quickly the oscillator back to the vacuum state. This method can be applied only for two coupled modes. In order to extend it to the case with more coupled modes, the authors use optimal control, in order to design the control inputs, which can quickly achieve the same tasks as before. The simulation predictions for the system are experimentally confirmed using a circuit quantum electrodynamics system.

We believe that the present work is significant for the field of quantum dynamics and control because it presents very interesting theoretical methods which are also experimentally applied in circuit quantum electrodynamics, a promising platform for quantum computation. The article is well written and scientifically sound, while the methods employed are explained in detail in the supplementary material. For these reasons we believe that the article warrants publication in Nature Communications, after the authors take into account the following observations:

We thank the reviewer for the careful reading of our manuscript and the constructive comments and suggestions that have helped improve our paper's quality. The following are point-by-point replies to specific issues raised.

1. They should explain what are the IQ initials appearing in the IQ trajectories displayed in several figures.

We have added black crosses in each IQ figure indicating the trajectory starting points and included an additional explanation of the IQ (In-phase, Quadrature) initials in the caption of Fig. 2.

2. The authors need to cite some related works. Specifically, the following article where numerical optimal control is used to find inputs in order to cool a noisy (open) quantum harmonic oscillator, and in the long time limit the system can be driven also along a decoherence free path:

- D. Stefanatos, Optimal efficiency of a noisy quantum heat engine, Phys. Rev. E 90, 012119 (2014).

We have added this citation as a quantum thermodynamics application in the last paragraph of the Discussion section.

Also, some articles where optimal control is used to obtain piecewise constant pulses, like those in the paper, which can be used to cool one of two coupled quantum oscillators:

- X. Wang, S. Vinjanampathy, F. W. Strauch, and K. Jacobs, Ultraefficient cooling of resonators: beating

sideband cooling with quantum control, Phys. Rev. Lett. 107, 177204 (2011).

- S. Machnes, J. Cerrillo, M. Aspelmeyer, W. Wieczorek, M. B. Plenio, and A. Retzker, Pulsed laser cooling for cavity optomechanical resonators, Phys. Rev. Lett. 108, 153601 (2012).
- D. Stefanatos, Fast cavity optomechanical cooling, Automatica 73, 71-75 (2016).

We have added these citations as possible applications in the optomechanical fields in the last paragraph of the Discussion section.

3. Line 13, there is an unnecessary “in” after “including”. Also, in Ref. [15] it is “Jarzynski”.

Corrected.

Referee #2:

In their manuscript "Shortcuts to Adiabaticity for Open Systems in Circuit Quantum Electrodynamics" the authors experimentally demonstrate a shortcut to adiabaticity for a system consisting of two transmon qubits and an oscillator. They propose a technique to perform the shortcut for an open quantum system as opposed to a closed system. They address an important experimental problem of designing STA protocols that can simultaneously suppress diabatic transitions in the presence of multiple modes. The multi-mode suppression is achieved using optimal control theory approach.

We thank the referee for the summary and recognition of the contribution of this work. To clarify, we should point out is that our system consists of one transmon qubit and two oscillators (one readout resonator, one Purcell filter).

I found the paper mostly well written, and the experiments to be well done. However, in order to be of interest to the broad audience of Nature Communications, I believe significant improvements to the paper should be done. For example, demonstrating that the proposed STA protocol can be used to improve qubit readout fidelity could be one way of adding value. In its current form, the results seem incompletely and would be more suitable for a specialized journal.

We are grateful to the reviewer for the many constructive comments and helpful feedback. We address specific comments below and believe that our updated manuscript now more clearly demonstrates the practical value of our approach and should be of interest to a broader audience now.

1. Fig. 4 is only introduced in the discussion section. The related discussion should be either moved to the results section or the supplementary material.

We have moved this part to the end of the MMOC section, as controlling unwanted excitations (or the phase transition of the non-linear cavity) is an important consideration worth mentioning in the main text when the MMOC is applied to a non-linear system.

2. In order to be able to assess the performance of the presented MMOC approach, it would have been

interesting to see it compared to other known strategies for optimizing resonator pulse shapes, such as CLEAR demonstrated in ref [41] of the paper or just a rectangular pulse.

We separate this comparison into two parts, regarding the reset and the charging processes, respectively.

Reset Process

(i) Comparison with rectangular pulse

A comparison between MMOC and a rectangular pulse is given in Fig. 3 of the main text, although we realize that may not have been apparent in the way it was described. More specifically, in Fig. 3(b,c), at ~940 ns, the resonator was reset either by MMOC ("MMOC with reset") or by turning off a rectangular pulse ("MMOC without reset"). After 1030 ns, the output signal from the MMOC reset is significantly smaller than that from the corresponding rectangular pulse. We have added a sentence to the main text (lines 140-142) to clarify this.

(ii) Comparison with CLEAR

A direct experimental comparison between MMOC and CLEAR seems difficult due to a lack of degrees of freedom in the CLEAR method.

CLEAR makes use of a 2-segment pulse, with the duration of each segment fixed. By solving the damped oscillator model, they deduce the absolute amplitude of each segment. There are thus 2 total degrees of freedom for control, whereas the target has 4 degrees of freedom (the qubit contributes 2, and the steady complex IQ point contributes 2). This approach should thus not give a general solution, although it is possible that with "good" parameters, the 2-segment CLEAR pulse with real amplitudes works well in practice. This may be why the authors have to resort to numerical optimization to reduce the duration from 300 ns to 240 ns ($1.6 \cdot \kappa^{-1}$).

In contrast, with MMOC, the total pulse duration is 60 ns (or about $1 \cdot \kappa^{-1}$, Fig.3 of the main text). The pulse consists of 10 segments with complex amplitudes, giving 20 degrees of freedom of control, which is more than enough given a target degree of freedom of 8: 2 qubit* (2 complex resonator IQ + 2 complex filter IQ). MMOC thus has advantages over CLEAR in terms of total pulse duration as well as controllability.

Charging Process

(i) Comparison with rectangular pulse

We have performed an additional experiment to understand how MMOC can be used for qubit readout by applying MMOC to the rising edge of the readout pulse and comparing the performance with a rectangular pulse.

Our results (Supplementary Figs. 14 and 15) show that MMOC can improve readout fidelity for qubits in the 0 state. For qubits in the 1 state, we find that the additional time required for the MMOC pulse leads to slightly reduced readout fidelity, but this can be addressed by reducing the length of the MMOC pulse and improving the T1 time of the qubit. In addition, a Bayesian-based algorithm can be used to correct bit flips during readout and further improve the quality of MMOC readout compared with the traditional statistical analysis based on IQ points. This is ongoing work and will appear as a separate manuscript.

(ii) Comparison with CLEAR

For the charging process, neither CLEAR nor any other related papers discuss how their novel protocols can improve readout fidelity, emphasizing instead on cavity depletion. That said, such methods use pulses with varying amplitudes, and hence the output signal, which is the vector superposition of the input and the leakage of the system, will also vary significantly and make the collected IQ plot orderless thus degrading the readout. As MMOC has a shorter duration ($1/\kappa$), we can instead collect the IQ points after the pulse and only suffer slightly from T1 decay that occurs during the pulse.

What are the benefits of the presented scheme over other numerical optimization schemes?

We have added a comment to the Discussion section.

Compared with other ansatz-based numerical optimal control methods, which can be more time-consuming and prone to getting stuck in local minima, our MMOC approach is based on a solid initial analytical solution -- we believe this is a better ansatz -- that can be further optimized based on experimental data to mitigate error arising from initial parameters, nonlinearity, and control distortion. In addition, MMOC has enough degrees of freedom (compared with, say, CLEAR), which can be optimized to satisfy different user-defined objectives (e.g., minimizing the maximum pulse amplitude to avoid unwanted qubit excitations).

Is the optimized MMOC approach still a shortcut to adiabaticity? I would think that the adiabatic trajectory is no longer followed.

Our MMOC method can be regarded as one way to realize the end-state outcomes of general STA. Counter-diabetic driving is needed to fully realize the fast adiabatic following of the desired dynamics (Hermitian or non-Hermitian), and the required control is uniquely defined. However, if we only need to guarantee the initial and final states, there is additional freedom to choose the internal trajectory, which can be used to realize other aims (multi-mode control here). The relative value and attractiveness of both approaches then depend on the specific use case under consideration. The review paper Ref [1] is an excellent reference for more strategies and the relation between STA and optimal control.

3. From the narrative it is not entirely clear why is the presence of the qubit important for the experiment. The application for readout optimization is only mentioned in the very end of the paper. This should be clarified. Similarly, in the abstract applications such as biophysics are given as an example, but the performed experiment seems more related to qubit readout optimization.

We have made some additions to the main text to clarify the presence of the qubit and connect the experimental setup to potential applications such as qubit readout fidelity and cycle time.

Regarding the connection to biophysics: while our results are indeed focused on qubit readout optimization, the extension of STA itself to open systems has potentially widespread applications (see, for instance, Iram, S. et al. Controlling the speed and trajectory of evolution with counterdiabetic driving. *Nature Physics* 17, 135–142 (2021)), and we hope to draw attention to the connections to other fields, so that value from this technique can be applied to other areas.

4. How significant impact did the inclusion of κ into the counterdiabatic driving protocol have? It would be interesting to see if the counterdiabatic driving designed for open quantum system works significantly better than the approximation derived for closed system.

We have performed an additional experiment to compare the open system CD drive to the closed system (κ set to 0). Because κ and the detuning have the same magnitude, only the open CD will accelerate the equilibrium. Our results are given in Supplementary Figure 16. As some parameters are different from those used in the main text, we have kept this separate from Fig. 2 of the main text.

5. Font size in the figures is quite small. I would suggest increasing them for better readability.

Figures and corresponding font sizes have been increased.

Reviewers' Comments:

Reviewer #2:

Remarks to the Author:

I thank the authors for carefully answering to all my questions. While I still think that the article would be better suited for a more specialized journal, I found that the additional analysis by the authors successfully highlight the merits of their method.

The work itself is definitely a fine addition to the literature of shortcuts to adiabaticity and scientifically sound. I leave it to the editors discretion whether these merits warrant the publication in Nature comms.